# Poly(alizarin red S) modified glassy carbon electrode for square wave adsorptive stripping voltammetric determination of metronidazole in tablet formulation

**Mulugeta Dawit[1], Mahilet Turbale[2], Amsalu Moges[3], Meareg Amare[4]***

**1** Debre Tabor University, Debra Tabor, Ethiopia, **2** Samara University, Semera, Ethiopia, **3** Debre Markos University, Debre Markos, Ethiopia, **4** Bahir Dar University, Bahir Dar, Ethiopia

* amaremeareg@yahoo.com

**Data Availability Statement:** All relevant data are within the manuscript.

**Funding:** The authors received no specific funding for this work.

## Abstract

Potentiodynamically fabricated poly(alizarin red s) modified GCE was characterized using CV and EIS techniques. In contrast to the cyclic voltammetric response of the unmodified GCE for metronidazole, an irreversible reduction peak with three-folds of current enhancement and reduced overpotential at the poly(alizarin red s) modified GCE showed the catalytic effect of the modifier towards reduction of metronidazole. While observed peak potential shift with increasing pH (4.0–10.0) indicated the involvement of protons during the reduction of metronidazole, peak potential shift with scan rate (20–300 mV s$^{-1}$) confirmed the irreversibility of the reduction reaction of metronidazole at the modified GCE. A better correlation for the dependence of peak current on scan rate ($r^2 = 0.9883$) than on square root of scan rate ($r^2 = 0.9740$) supplemented by slope value of 0.38 for plot of log(current) versus log(scan rate) indicated the reduction reaction of metronidazole at the surface of the modified electrode was predominantly adsorption controlled. Under the optimized method and solution parameters, reductive current response of tablet sample showed linear dependence on spiked standard concentration in a wide range (0–125 µM) with excellent determination coefficient $r^2$, LoD and LoQ of 0.9991, 0.38, and 1.25 µM, respectively. Spike recovery of 97.9% and interference recovery of 96.2–97.5% in the presence of 21.28 and 31.92 µM of uric acid and ascorbic acid validated the applicability of the present method for determination of metronidazole in tablet formulation. The metronidazole content of the tested tablet formulation using standard addition method was found to be 97.6% of what is claimed by the tablet manufacturer making the developed method an excellent potential candidate for its applicability to determine metronidazole in real samples with complex matrix.

## 1. Introduction

Metronidazole (*2-methyl-5-nitroimidazole-1-ethanol*) which is used for treatment of infections caused by anaerobic bacteria (*Bacteroides*, *Fusobacterium*, *Campylobacterium*, and

**Competing interests:** The authors have declared that no competing interests exist.

*Clostridium*), protozoa (*Trichomonas*, *Treponema*, and *Histomonas*), and amoeba belongs to the nitroimidazole drug family [1]. It kills or inhibits majority of anaerobic bacteria at its concentration in serum in the range of 2–8 mg/mL [2]. Due to its antimicrobial activity, rapid bacterial killing, good tissue penetration, low cost, and limited adverse effects; metronidazole (MTZ) is the drug of choice for prevention and treatment of patients with Crohn's disease and ulcerative colitis [3, 4].

The pharmacokinetic and pharmacodynamic properties of the drug available as oral, intravenous, vaginal, and topical formulations are favorable. In accordance with the international guidelines, MTZ is also a component of multidrug regimens (e.g., in combination with omeprazole, rabeprazole, and amoxicillin) for therapy of *Helicobacter pylori* infections, which is a major cause of gastritis and a risk factor for stomach cancer [3]. For its low-price, and effective veterinary drug, MTZ (Scheme 1) is also used as the growth promoter in agriculture, aquaculture, livestock and bee-keeping.

Development of leucopenia, neutropenia, increased risk of peripheral neuropathy, and toxicity on the central nervous system are however concomitants of high doses and long-term systemic treatment with MTZ. In clinical studies where high doses of MTZ were used during radiation treatment for cancer, an overdose of the drug was reported to increase the risk of seizures or nerve problems in the hands and feet [5]. The medication is most likely to cause problems in the case of overdose when it is taken by mouth or by intravenous (IV), rather than applied to the skin or used vaginally [6]. Because of its potential toxicity in human health, its use in food and animal feeds have been prohibited [7]. Hence, it is very important to have a sensitive, selective and accurate method to monitor level of MTZ in biological, pharmaceutical and environmental samples.

Capillary electrophoresis [8], high performance liquid chromatography [9], titrimetry [10], and spectrophotometry [11] are among the commonly reported methods for determination of MTZ in pharmaceutical samples. However, most of these methods are known to have limitations in simplicity, cost-effectiveness, easy access, and environmental friendly [12]. Because of their inherent advantages of simplicity, ease of miniaturization, high degree of accuracy, precision, sensitivity, selectivity, and relatively low cost, electrochemical methods are promising alternatives for determination of electroactive species including MTZ [13, 14].

Attempts have been reported on the electrochemical determination of MTZ in pharmaceutical and clinical matrices using carbon paste electrode [15], activated glassy carbon electrode [16], α-cyclodextrin/CPE [17], ZnCo-MOF/GCE [18] and Ag/Au/Nafion/GCE [19]. Although the reported methods are sensitive with detection to a nanomolar level, they have still limitations associated with environmental, availability, and preparation complexity issues. Thus, development of a simple, cost effective, and sensitive method for determination of MTZ in real samples including pharmaceutical formulations is still vital.

Compared to metal electrodes, glassy carbon electrode (GCE) is widely used due to its biocompatibility with tissue, low residual current over a wide potential range, and minimal

**Scheme 1. Chemical structure of MTZ.**

propensity to show deteriorated response as a result of electrode fouling [20]. Modifying its surface with a material that improves its activity towards the analyte of interest further increases its applicability [21, 22]. Due to its stability, reproducibility, increase in active sites, homogeneity in electrochemical deposition, and strong adherence to electrode surface, polymer-modified electrode (PME) has received attention in the area [23]. It has been demonstrated in many reports that electrodes modified with redox active polymer films including dye and dyestuffs show excellent stability, reproducibility, and homogeneity [24, 25]. Alizarin Red S (ARS) is one of the common electroactive dyes recently reported for fabrication of electroactive polymer film modified electrodes in the field of electrochemical sensors development [26–28].

Therefore, the present study describes the application of an accurate, precise, and selective standard addition method based on poly(Alizarin red S) modified glassy carbon electrode (PARS/GCE) for determination of MTZ in tablet sample, which to the best of our knowledge has not been communicated previously for the same.

## 2. Experimental part

### 2.1. Chemicals and apparatus

Standard metronidazole (100%, Emmelen Biotech Pharmaceuticals Limited), metronidazole tablet of Ethiopian Pharmaceuticals factory (EPHARM) brand, Alizarin Red S (97%, Samir Tech-chem.2TD), $Na_2HPO_4$ (99%) and $NaH_2PO_4$ (97%) (Sisco Research Laboratories Pvt. Ltd), HCl (35.4%, Lobal chemie), NaOH (97%, Blulux Laboratories Ltd), ascorbic acid (99%, Blulux Laboratories Ltd), $K_4[Fe(CN)_6]$, $K_3[Fe(CN)_6]$, and KCl were used. All chemicals were of analytical grade that they were used without prior purification. Distilled water was used throughout the work.

CHI760E Electrochemical Workstation (CHI Instruments, Austin, Texas, USA), pH meter (Adawa model AD800), electronic balance (Denver Instrument), and refrigerator were used for electrochemical data acquisition, pH adjustment, weighing mass and preserving sample, respectively.

### 2.2. Electrode preparation

Deposition of PARS polymer film at the surface of glassy carbon electrode was performed following reported procedure with minor modification [25]. Briefly: the glassy carbon electrode (3 mm diameter) was polished using aqueous slurries of alumina (1.0, 0.3, and 0.05 μm size successively) on polishing cloth, and then thoroughly rinsed with distilled water. Electropolymerization of ARS was performed by cyclic voltammetric scanning of GCE in pH 3 phosphate buffer solution containing 1.0 mM ARS in the range of -0.2 to +1.8 V for 15 cycles at a scan rate of 100 mV s$^{-1}$. The fabricated PARS/GCE was then rinsed with distilled water and stabilized by scanning between -0.8 to +0.8 V at 100 mV s$^{-1}$ in monomer free 0.5 M $H_2SO_4$ until a stable voltammogram was obtained. The stabilized PARS/GCE was allowed to dry in air before use.

### 2.3. Electrochemical measurements

A conventional three electrode cell, consisting of unmodified GCE or PARS/GCE as working electrode, Ag/AgCl (3 M KCl) as reference, and a platinum coil as auxiliary electrode was used for electrochemical measurements. All potentials mentioned in this paper refer to the Ag/AgCl reference electrode.

While cyclic voltammetry was used to deposit polymer film on the surface of GCE, characterize the polymer film, and investigate behavior of MTZ at the modified electrode, effect of scan rate and solution pH on both peak current and peak potential; square wave stripping voltammetry was employed for the quantitative analysis of MTZ in tablet sample.

### 2.4. Preparation of standard solutions

Phosphate buffer solutions (PBS) in the pH range 4.0–10.0 were prepared from equi-molar (0.1 M) mixture of disodium hydrogen phosphate and sodium dihydrogen phosphate. The pH of the solutions was adjusted using 0.1 M of NaOH and HCl solutions as required.

While 5.0 mM stock solution of MTZ was prepared by dissolving 0.0856 g of standard MTZ in 100 mL of distilled water, working solution of 1.0 mM concentration was prepared from the stock solution using pH 7.0 PBS.

### 2.5. Tablet sample preparation

MTZ tablet sample was prepared following reported procedure with minor modification [15]. MTZ tablets (all labelled as 250 mg per tablet) of Ethiopian Pharmaceuticals Factory (EPHARM) brand were purchased from a pharmacy in Bahir Dar city, Ethiopia. Randomly selected five tablets were accurately weighed and ground using mortar and pestle. An adequate amount of this powder (170 mg) corresponding to claimed concentration of 10.0 mM was transferred into a 100 mL volumetric flask and filled to the mark with distilled water. Insoluble residue was discarded by filtration and the volume was readjusted to the mark with distilled water. An intermediate tablet sample solution of 0.5 mM in pH 7.0 was prepared from the tablet stock solution. Tablet sample solutions with claimed concentration of 21.8 $\mu$M in pH 7.0 PBS were prepared from the intermediate tablet sample solution and used for determination of MTZ level in the tablet sample, spike recovery and interference recovery analysis.

## 3. Results and discussion

### 3.1. Fabrication of PARS/GCE

PARS/GCE was fabricated potentiodynamically by scanning the potential of polished glassy carbon electrode in 1.0 mM alizarin red S monomer solution between -0.2 and +1.8 V for 15 cycles at a scan rate of 100 mV s$^{-1}$ [26] (Fig 1). During the cyclic voltammetric electopolymerization process of ARS on the surface of GCE (Fig 1), an oxidative peak (peak-1) and reductive peak (peak-2) appeared at +1.48 and + 0.54 V, respectively. In agreement to the trend in reported literature [26], the peak current of the two peaks increased with scan cycles showing the deposition of a polymer film on the surface of the GCE.

As can be seen from the voltammograms of the unmodified (curve A) and modified (curve B) GCEs in a monomer free 0.5 M H$_2$SO$_4$ (Inset of Fig 1), while the peaks designated by "$a$" and "$a^*$" at the unmodified and modified electrodes, respectively are ascribed for molecular oxygen reduction, the two distinct redox couples "$b$"-"$b^*$" (0.38, 0.31 V) and "$c$"-"$c^*$" (-0.25, -0.28 V) that appeared only at the modified electrode confirmed deposition of a redox active polymer film at the surface of the glassy carbon electrode. In contrast to the reduction potential value (-0.55 V) for molecular oxygen at the unmodified electrode "$a$", appearance of the reductive peak at lower potential (-0.40 V) at the polymer modified electrode "$a^*$" indicated sort of catalytic property of the modified electrode towards oxygen reduction and hence confirmation of surface modification.

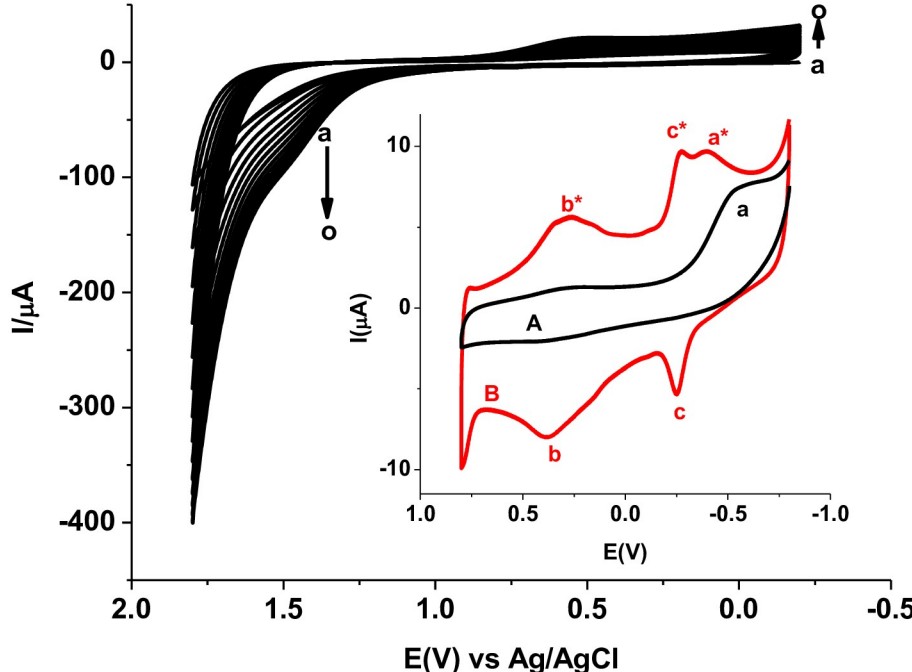

**Fig 1. Cyclic voltammograms of GCE in pH 3.0 PBS containing 1.0 mM ARS scanned between -0.2 V to +1.8 V at 100 mV/s for 15 cycles.** Inset: cyclic voltammograms of (A) bare GCE and (B) stabilized PARS/GCE in a monomer free 0.5 M $H_2SO_4$ scanned between -0.8 and +0.8 V at 100 mVs$^{-1}$.

## 3.2. Characterization of PARS/GCE

The modification of the surface of the GCE by a polymer film of PARS was further evidenced by results obtained using two techniques; cyclic voltammetry (CV) and electrochemical impedance spectroscopy (EIS) taking $[Fe(CN)_6]^{3-/4-}$ as a probe.

**3.2.1. Characterization by CV.** Appearance of couple of redox peaks in opposite scan directions for $[Fe(CN)_6]^{3-/4-}$ at both unmodified and polymer modified glassy carbon electrodes is characteristic of the probe (Fig 2). In contrast to the peak-peak separation ("*a*"–"*b*") of (ΔE 340 mV) at the unmodified electrode, lower peak-peak ("*a*\*"–"*b*\*") potential separation (ΔEp 104 mV) added with enhanced peak current for the probe at the polymer modified electrode indicated the catalytic property of the polymer film towards the probe and hence confirmed the modification of the electrode surface. The observed catalytic effect of the modifier might be due to an increased conductivity of the surface and hence facilitating electron exchange between the electrode and the probe at the electrode-solution interface and/or increased effective surface area of the modified electrode.

**3.2.2. Characterization by EIS.** Fig 3 presents the Nyquist plot of both the unmodified and modified electrodes in $[Fe(CN)_6]^{3-/4-}$. As can be seen from the figure, both electrodes showed a Nyquist curve with a semi-circle of different diameter in the high frequency region and a linear line in the low frequency region. In contrast to the unmodified electrode, a semi-circle of smaller diameter at the polymer modified electrode indicated lower charge transfer resistance and hence fast electron exchange between the polymer surface and the electroactive probe at the electrode-solution interface. The lower peak-peak separation observed for the probe at the PARS/GCE in Fig 2 can thus be ascribed to the improved conductivity of the surface of the modified electrode.

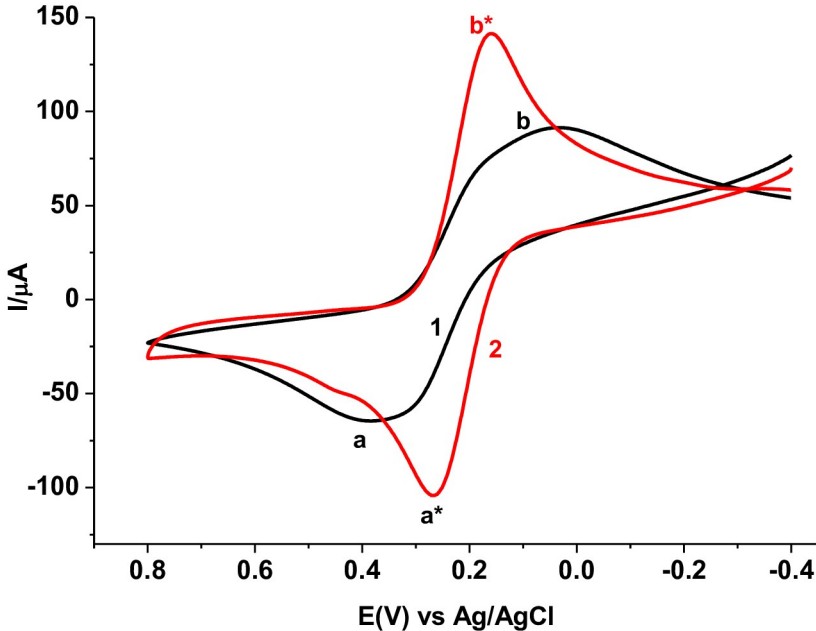

**Fig 2. Cyclic voltammograms of (1) bare GCE and (2) PARS/GCE in pH 7.0 PBS containing 10.0 mM [Fe(CN)$_6$]$^{3-/}$$^{4-}$ and 0.1 M KCl. Scan rate 100 mV s$^{-1}$.**

The presumed RC-parameters (R$_s$, $R_{ct}$ and $C_{dl}$) for the studied electrodes as calculated using eq (1) from the respective semi-circles is summarized in Table 1.

$$C_{dl} \;=\; \frac{1}{2\pi \, R_{ct} f_{max}} \tag{1}$$

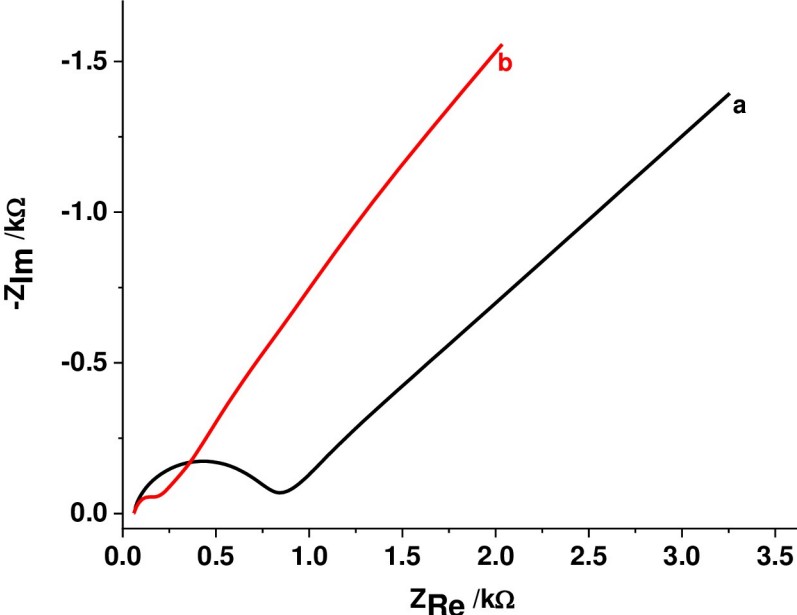

**Fig 3. Nyquist plots for (a) bare GCE and (b) PARS/GCE in pH 7.0 PBS containing 10.0 mM [Fe(CN)$_6$]$^{3-/4-}$ and 0.1 M KCl at frequency range: 0.01–100,000 Hz, applied potential: +0.23 V, and amplitude: 0.01 V.**

**Table 1. Summary of the calculated values of selected RC-elements.**

| Electrode type | $R_s$ (kΩ) | $R_{ct}$ (kΩ) | $C_{dl}$ (F) |
|---|---|---|---|
| Bare GCE | 73 | 926 | $4.32 \times 10^{-11}$ |
| PARS/GCE | 73 | 278 | $2 \times 10^{-9}$ |

where $f_{max}$, is the frequency (Hz) corresponding to the maximum value of $-Z''$ at the semi-circle, $C_{dl}$ is the double layer capacitance, $R_{ct}$ is the charge transfer resistance given by the diameter of the semi-circle, and $R_s$ is solution resistance given by the x-axis value corresponding to the semi-circle at maximum frequency.

While higher double layer capacitance ($C_{dl}$ $2 \times 10^{-9}$) of the modified electrode indicates deposition of a certain material on the surface of the electrode, lower charge transfer resistance value ($R_{ct}$ 278) indicated surface modification by a more conductive material which is responsible to lower the redox potentials of an electroactive species. Therefore, the results obtained from the EIS data are in support of the effects of the modified electrode towards $[Fe(CN)_6]^{3-/4-}$ probe confirming the modification of the surface of the electrode by an intrinsically conducting polymer film.

## 3.3. Cyclic voltammetric investigation of MTZ at PARS/GCE

**3.3.1. Electrochemical behavior of MTZ at PARS/GCE.** The electrochemical behavior of MTZ at the PARS/GCE, and effects of scan rate ($v$) and pH of supporting electrolyte on both the peak current (Ip) and peak potential (Ep) of MTZ at PARS/GCE were studied using CV. The reductive peaks "1" and "1*" at the unmodified and polymer modified electrodes, respectively both in the absence of MTZ (curves a, and b of Fig 4) are ascribed to reduction of molecular oxygen showing potential interference of oxygen during analysis of MTZ unless the sample solution is bubbled with nitrogen or sort of blank subtraction is made.

Although of differing intensities, a single well resolved peak in the reduction scan direction without peak in the reverse scan direction at both electrodes for MTZ (curves c and d of Fig 4) at both electrodes indicated the irreversibility of the reduction reaction of MTZ at both electrodes. In contrast to the subtracted for blank reductive peak (Ep -741 mV) at the unmodified electrode (curve a of Inset), appearance of the same peak at a reduced potential (Ep -670 mV) with over three-folds of reductive peak current at the polymer modified electrode (curve b of Inset) indicated catalytic property of the polymer film towards reduction of MTZ. The observed catalytic effect of the modifier towards MTZ reduction might be accounted for the improved conductivity of the polymer film as obtained from the EIS data, and/or increased effective surface area of the modified electrode.

**3.3.2. Effect of pH on Ip and Ep of MTZ.** Investigation of the effect of pH of supporting electrolyte in electrochemical studies helps to assess involvement of protons during the reaction, proton to electron ratio, and possible interaction between the electroactive species and the electrode surface thereby proposing possible reaction mechanism.

In this study, cyclic voltammograms of PARS/GCE in PBS of various pH (4.0–8.0), all containing same concentration of MTZ, were recoded (Fig 5A). While the observable potential shift in the negative direction with increasing pH (a-g of Fig 5A) showed participation of protons in the reduction of MTZ at the surface of the modified electrode, a regression equation with slope value of 43.1 mV for the linear dependence of Ep on pH values (curve b for Fig 5B), which is close to the ideal value of 59 mV [29], showed the ratio of protons to electrons is 1:1.

Moreover, the observed increase in reductive peak current with pH from pH 4.0 to 7.0 which then declined beyond pH 7.0 (curve a of Fig 5B) showed pH 7.0 as the optimum value

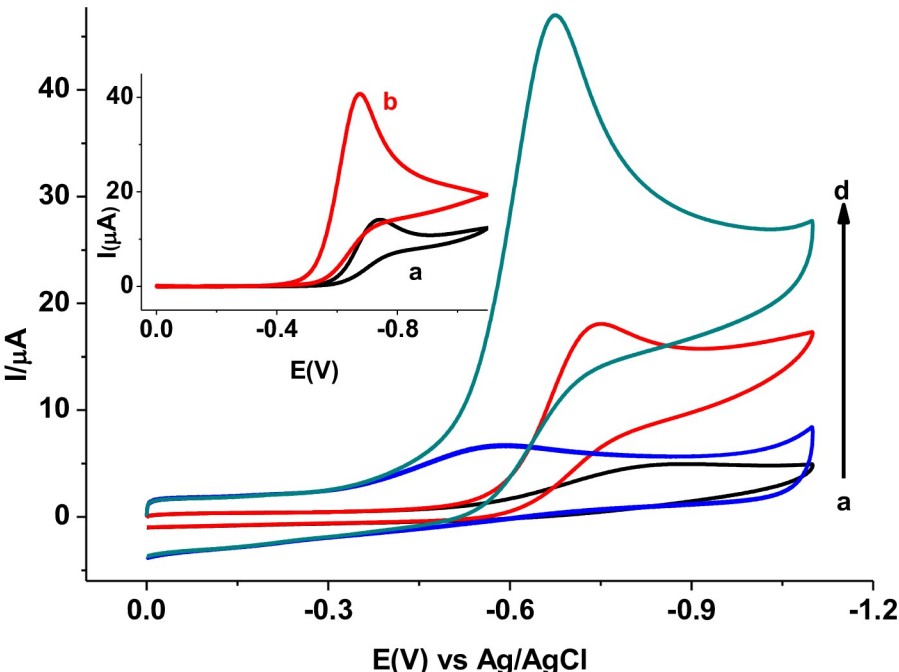

**Fig 4. Cyclic voltammograms of bare GCE (a & c) and PARS/GCE (b & d) in the absence (a & b) and presence (c & d) of 1.0 mM MTZ in pH 7.0 PBS at scan rate 100 mV s$^{-1}$.** Inset: corrected for blank cyclic voltammograms of bare (a) and PARS/GCE (b).

which is in agreement with literature [11]. The fact that pKa of MTZ is 2.6 while pKa of ARS are 5.49 and 10.85; the observed increasing current response with pH from 4.0 to 7.0 might be accounted for the possible electrostatic attraction between the negatively charged MTZ (pH > 2.6) and still positively charged polymer film whereas the decreasing trend at pH beyond 7.0 be due to the repulsive force exhibited between the negatively charged MTZ and polymer film of the electrode (pH >> 5.49). A reaction mechanism which is in agreement with reported literature [14], was proposed for the irreversible reduction of MTZ at the PARS/GCE (Scheme 2).

**3.3.3. Effect of $v$ on Ip and Ep.** The effect of scan rate on the reduction peak potential and peak current of MTZ at PARS/GCE was studied in the range of 20–300 mV s$^{-1}$. Observed peak potential shift in the negative potential direction with increasing scan rate (Fig 6) confirmed the irreversibility of the reduction reaction of MTZ at PARS/GCE [30].

While determination coefficient values ($R^2$) for plot of Ip *versus v* (Fig 7B) and Ip *versus v*$^{1/2}$ (Fig 7A) of 0.9883 and 0.9740, respectively indicated mixed diffusion-adsorption controlled with predominantly adsorption controlled mechanism [31], a slope value of 0.38, which is less than the ideal value of 0.5 for plot of log (Ip) *versus* log($v$) (Fig 8), further confirmed adsorption predominant reaction [32, 33].

## 3.4. Square wave voltammetric investigation of MTZ at PARS/GCE

To further elaborate the catalytic effect of the PARS/GCE towards reduction of MTZ, square wave voltammograms (SWVs) were recorded (Fig 9). In agreement with our CV results, an extremely sharp peak (-674 mV) with three folds of current response at PARS/GCE (curve b) relative to the peak (-740 mV) at the unmodified GCE (curve a) confirmed the catalytic

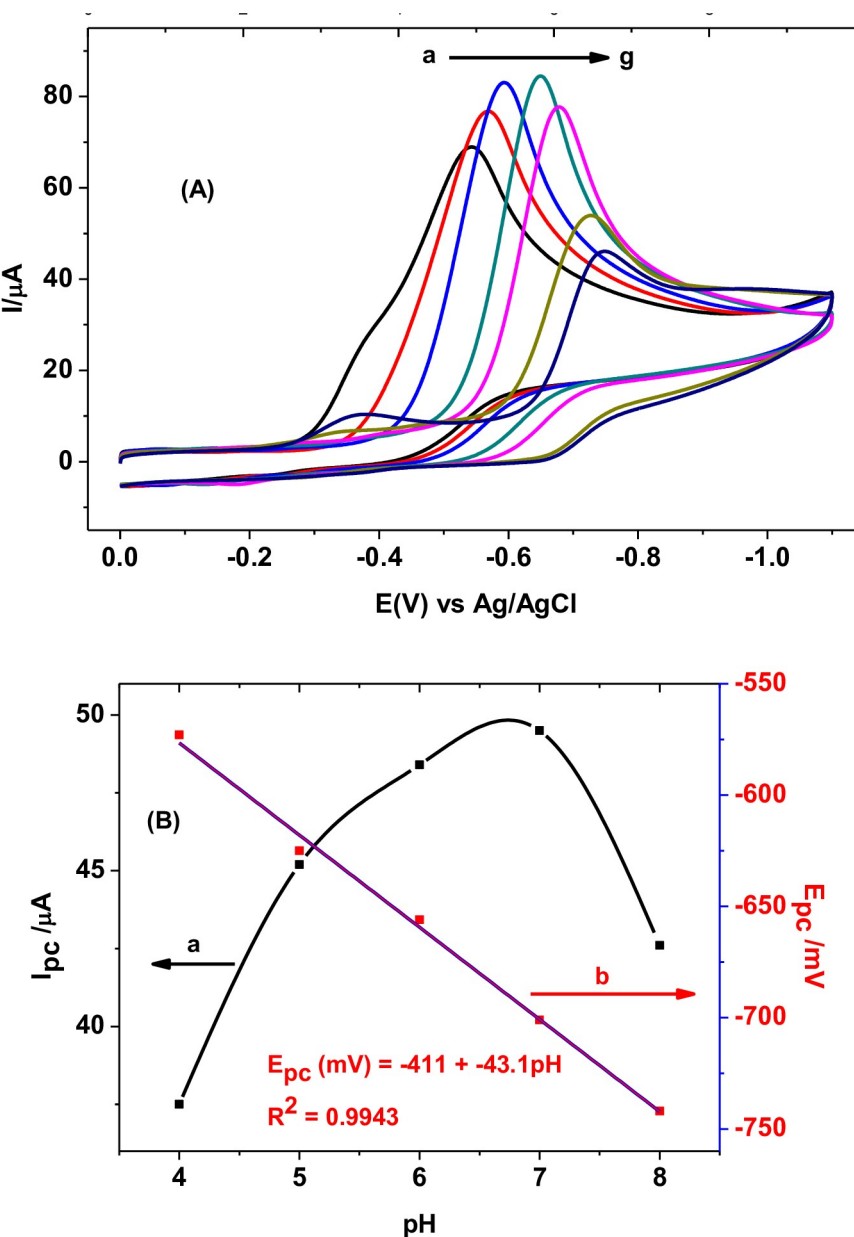

**Fig 5.** (A) Cyclic voltammograms of 1.0 mM MTZ at PARS/GCE in PBS of different pH values (a–g: 4.0, 5.0, 6.0, 7.0, 8.0, 9.0, and 10.0, respectively) at scan rate of 100 mV s⁻¹. (B) Plot of reductive (a) Ip and (b) Ep *versus* pH for 1.0 mM MTZ in PBS values at PARS/GCE. Scan rate: 100 mV s⁻¹.

property of the polymer modified electrode over the unmodified electrode towards reduction of MTZ.

**3.4.1. Effect of accumulation time and accumulation potential.** As investigation of the effect of scan rate on the peak current revealed a reaction predominantly controlled by adsorption processes, an attempt was made to check the effect of accumulation parameters on the current response of the modified electrode. While the influence of accumulation potential ($E_{acc}$) was investigated over the potential range -300 to -500 mV at an accumulation time ($t_{acc}$)

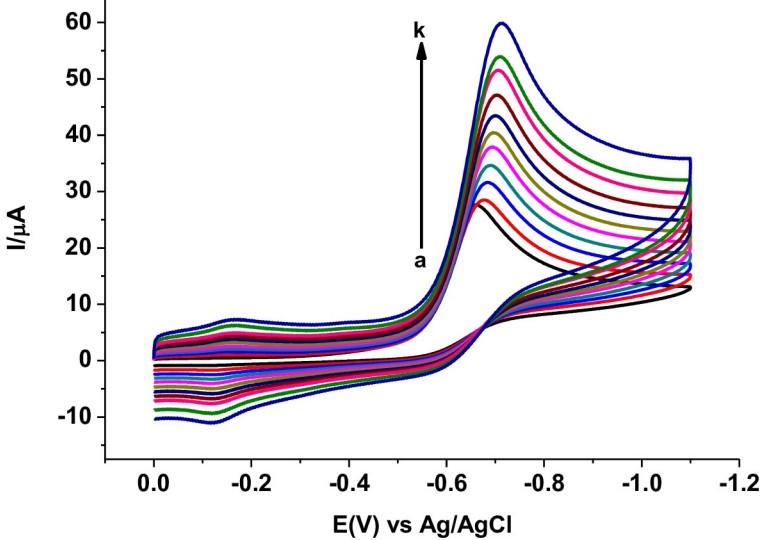

**Scheme 2. The proposed reaction mechanism of MTZ at PARS/GCE.**

of 10 s (figure not shown), effect of accumulation time was also checked by varying the time from 5 through 60 s.

The reductive peak current increased dramatically with $E_{acc}$ from -300 to -450 mV which then leveled off at a potentials beyond it making -450 mV the optimum value. The current response of the modified electrode at $E_{acc}$ of -450 mV increased with $t_{acc}$ throughout the studied time range but with a decreasing sensitivity (figure not shown). Thus, as a compromise between the current increment and analysis time, an accumulation time of 20 s was selected. Therefore, square wave adsorptive cathodic stripping voltammetry (SWAdCSV) with $E_{acc}$ and $t_{acc}$ values of -450 mV, and 20 s, respectively was employed for determination of the MTZ content in a tablet sample.

**3.4.2. Standard addition method of calibration.** As the matrix in a tablet is complex, standard addition method of analysis is recommended to compensate for the possible errors associated due to matrix. Fig 10 presents the back ground corrected SWAdCSV of tablet sample spiked with various concentrations of standard MTZ at PARS/GCE. Under the optimized solution and method parameters, mean corrected for blank SWAdCSV peak current of MTZ spiked tablet sample at PARS/GCE was linearly proportional to the spiked MTZ concentration in the range of 0–125 μM with linear regression equation and determination coefficient of Ip/μA = 7.66±0.29 + 0.36±0.00 [MTZ] $\mu$M and $R^2$ = 0.9991, respectively (Inset of Fig 10). The calculated method limit of detection (LoD = 3s/m) and limit of quantification (LoQ = 10s/m)

**Fig 6. Cyclic voltammograms of 1.0 mM MTZ in pH 7.0 PBS at PARS/GCE at various scan rates (a–k: 20, 40, 60, 80, 100, 125, 150, 175, 200, 250, and 300 mV s$^{-1}$, resp.).**

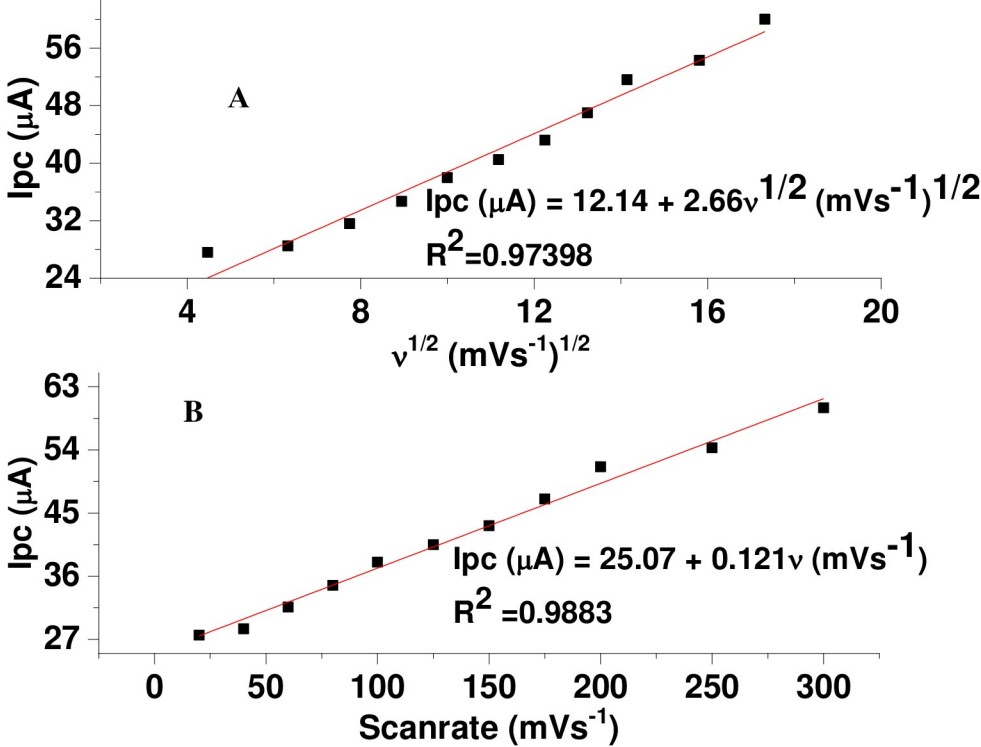

**Fig 7. Plot of Ip *versus* (a) $v^{1/2}$ and (b) $v$ for 1.0 mM MTZ in pH 7.0 PBS at PARS/GCE.**

were 0.38 μM and 1.25 μM, respectively; where s is blank standard deviation for *n* = 6) and *m* is the slope of the calibration graph [34]. The mean current results for the standard addition method were associated with errors in terms of RSD in the range 0.26% for 125 μM spiked standard MTZ to 10.01% for the unspiked sample confirming the stability of the modified electrode and precision of the method.

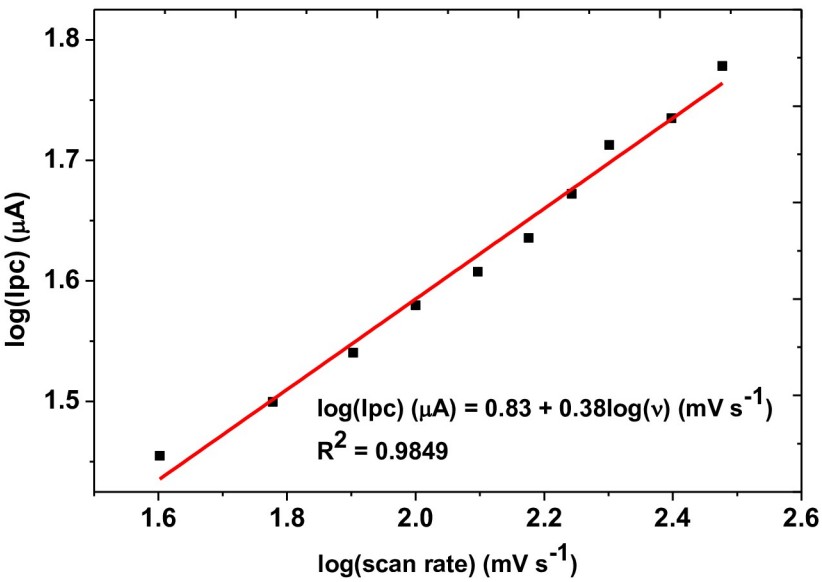

**Fig 8. Plot of log(Ip) *versus* log($v$) in a scan rate range of 40–300 mV s$^{-1}$.**

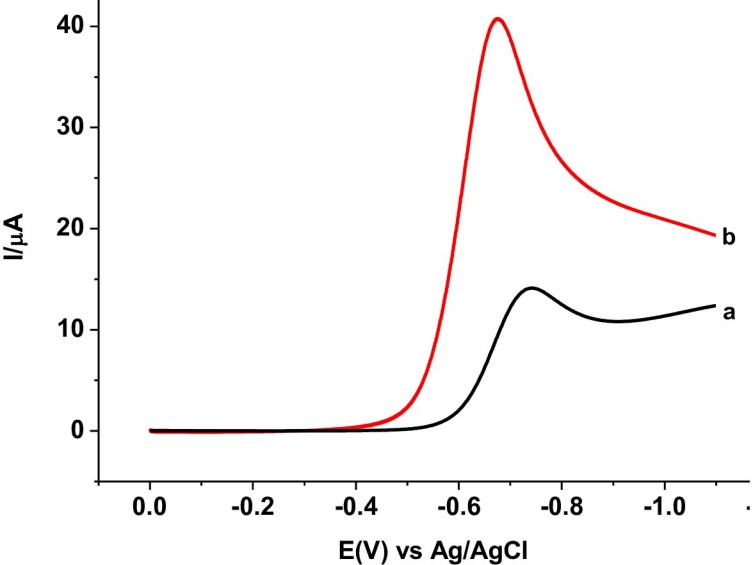

**Fig 9. Corrected for blank SWVs of 1.0 mM MTZ in pH 7.0 PBS at (a) bare GCE, and (b) PARS/GCE at step potential: 4 mV, amplitude: 25 mV and frequency: 15 Hz.**

**3.4.3. Determination of level of MTZ in a tablet sample.** The developed SWAdCSV method was used for determination of MTZ content in a tablet sample prepared as described under the experimental part. Besides the high precision of the method due to low RSD values (0.026–10.08%) observed for the standard addition calibration results, the applicability of the developed method for determination of MTZ in tablet sample was further validated using spike recovery and interference recovery results. The MTZ level in the studied tablet sample

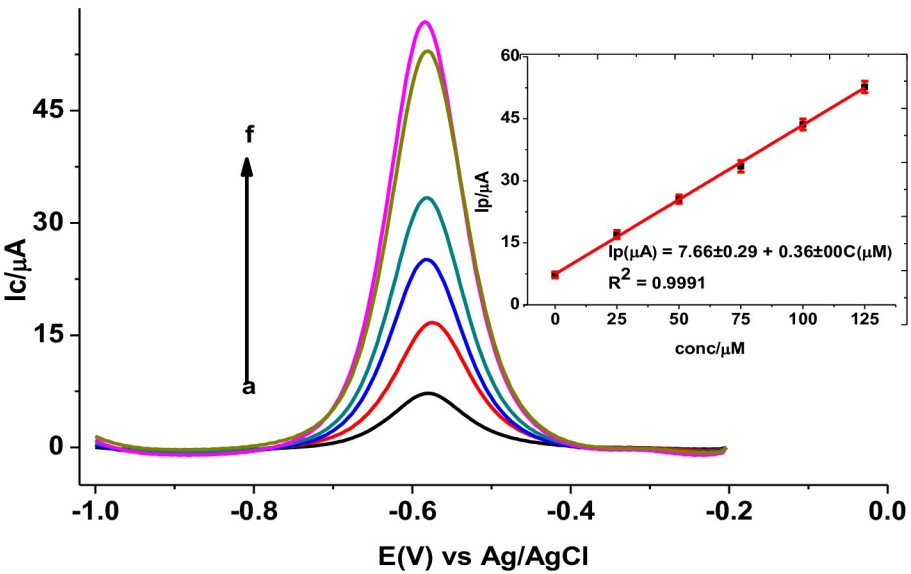

**Fig 10. Representative background corrected SWAdCSV of PARS/GCE in pH 7.0 PBS containing tablet sample spiked with various concentrations of standard MTZ (a–e: 0.0, 25.0, 50.0, 75.0, 100.0, and 125.0 μM, respectively).** Inset: plot of mean±SD (n = 3) of Ip *versus* spike concentration of MTZ.

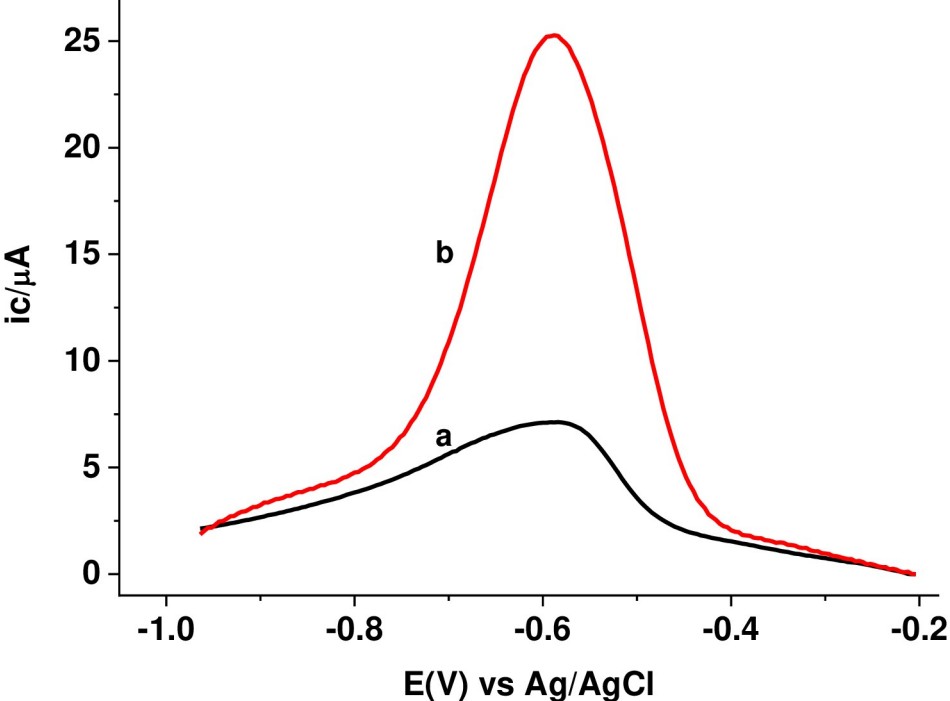

**Fig 11. SWAdCSV of EPHARM MTZ sample in PBS pH 7.0 spiked with 50 $\mu$M of standard MTZ.**

relative to its expected level according to the tablet label, spike recovery results (Fig 11) and interference recovery (Fig 12) results are summarized in Table 2.

As can be seen from the table, detection of an amount of MTZ in the tablet sample with an error of only 2.4% from the theoretical value indicates the extent of accuracy of the method. The accuracy of the method was further evaluated using the recovery result for a spiked amount of standard MTZ in the tablet sample. A spike recovery result of 97.88% from a tablet sample with a complex matrix composition still confirmed the accuracy of the method and hence its validity for drug determination in real samples.

Moreover, the selectivity of the method was evaluated by applying the method for determination of MTZ in a tablet formulation in the presence of 21.28 and 31.92 µM of ascorbic acid (AA), and uric acid (UA). Although presence of AA showed a decreasing level of MTZ with its increasing amount, the amount of MTZ detected still is in agreement with (96.2–97.5) the expected level according to the label. Detected values lower than the prescribed value may be due to the possible mass loss of MTZ during preparation or sort of degradation during storage, otherwise originally lower levels of MTZ in the tablets.

### 3.5. Comparison of the present method with previously reported methods

The performance of the present method was compared with reported methods in terms of their linear dynamic range, and limit of detection (Table 3). Although most of the reported methods [17–19] seem to have lower LoD and even wider linear range than the present study, the reported methods suffer from expensive electrode modifying material, and tedious modification steps. Thus, the present method using relatively cheap surface modifier and simple modifying step can be an excellent candidate for determination of MTZ in real samples.

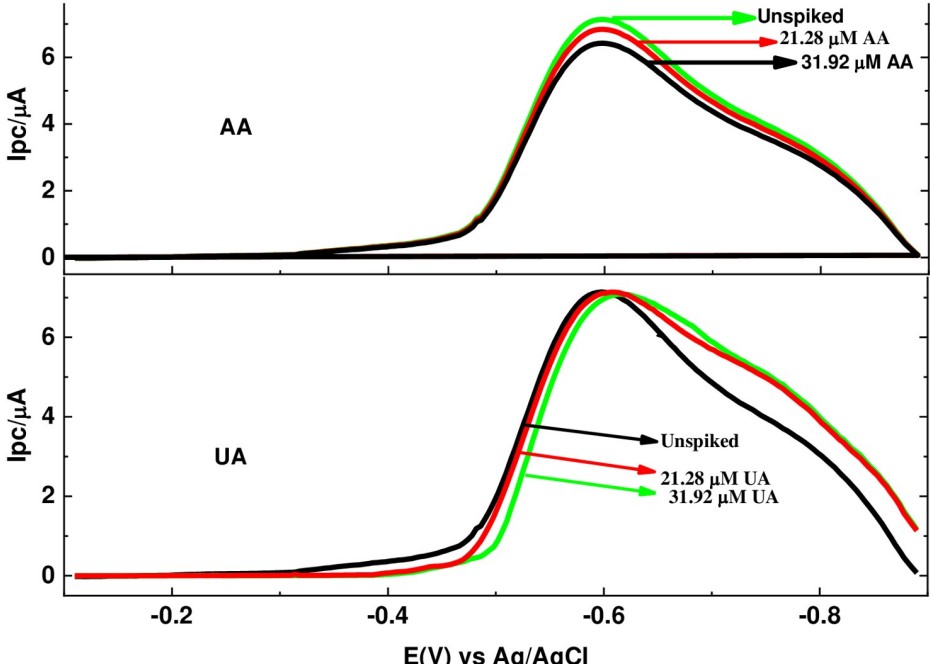

**Fig 12. SWAdCSV of pH 7.0 PBS containing EPHARM MTZ tablet solution in the presence of various concentrations of UA and AA.**

**Table 2. Summary results of level of MTZ in tablet sample, spike recovery, and interference recovery at 100 and 150% of selected potential interferents.**

| Purpose of analysis | Analyzed sample | Spiked MTZ (μM) | Added potential interferent (μM) | | Expected (μM) | Detected (μM) | Detected/recovery (%) |
|---|---|---|---|---|---|---|---|
| | | | AA | UA | | | |
| MTZ level in tablet sample | Tablet* | - - - | - - - | - - - | 21.80 | 21.28 | 97.6 |
| Spike recovery | Tablet** | 50.00 | - - - | - - - | 71.80 | 70.28 | 97.9 |
| Interference recovery | Tablet* | - - - | 21.28 | - - - | 21.80 | 21.08 | 96.7 |
| | Tablet* | | 31.92 | - - - | 21.80 | 20.96 | 96.2 |
| | Tablet* | | - - - | 21.28 | 21.80 | 21.26 | 97.5 |
| | Tablet* | | - - - | 31.92 | 21.80 | 21.23 | 97.4 |

* tablet sample prepared to be 21.80 μM as per to manufacturer's tablet label; UA uric acid; AA ascorbic acid

**Table 3. Comparison of several electrochemical methods for MTZ determination.**

| Electrode | Method | Sample analyzed | LoD (μM) | Linear range (μM) | Ref. |
|---|---|---|---|---|---|
| CPE | SWV | Tablet | 0.497 | 1–500 | [15] |
| Activated GCE | CV | Tablet | 1.1 | 2–600 | [16] |
| α-cyclodextrin/CPE | DPV | Tablet | 0.28 | 0.5–103.0 | [17] |
| ZnCo-MOF/GCE | LSV | Tablet | 0.017 | 0.05–100 | [18] |
| Ag/Au/Nafion/GCE | DPV | Serum | 0.059 | 100–1000 | [19] |
| PARS/GCE | SWAdCSV | Tablet | 0.375 | 25–125 | This work |

## 4. Conclusion

In this study, potentiodaynamically fabricated PARS/GCE was characterized using CV and EIS. While CV results showed the modification of the electrode surface by a redox active material, EIS results confirmed surface modification by a more conductive material. Electrochemical investigation of MTZ at the unmodified and modified electrodes revealed a peak in the reduction scan direction with no peak in the oxidation scan direction irreversibility of its reduction at both electrodes. In contrast to the unmodified glassy carbon electrode, reductive peak with about three folds enhanced peak current at the modified glassy carbon electrode indicated catalytic role of the modifier towards MTZ. While observed peak potential shift with increasing pH in the range 4.0–10.0 indicated the involvement of protons during the reduction of MTZ, the peak potential shift observed with scan rate in the range 20–300 mV/s confirmed the irreversibility of the reduction reaction of MTZ at the electrode. A better determination coefficient ($R^2$) for the dependence of peak current on the scan rate (0.9883) than on the square root of scan rate (0.9740) indicated reduction of MTZ was predominantly charge transfer controlled kinetics. The MTZ level of a target tablet sample labeled 250 mg/tablet using the developed method was found to be 244.04 mg/tablet which is with 2.34% error. Detected value lower than the prescribed value may be due to the possible mass loss of MTZ during preparation or sort of degradation during storage, otherwise originally lower levels of MTZ in the tablets. Recovery result of 97.9% for spiked standard MTZ in tablet sample and 96.2–97.5% recovery of MTZ in tablet sample in the presence of 21.28 and 31.92 μM of UA and AA, validated the applicability of the present method for determination of MTZ in real samples including tablet formulation.

## Author Contributions

**Conceptualization:** Meareg Amare.

**Data curation:** Mulugeta Dawit, Mahilet Turbale, Amsalu Moges.

**Formal analysis:** Mulugeta Dawit, Mahilet Turbale, Amsalu Moges.

**Methodology:** Meareg Amare.

**Resources:** Meareg Amare.

**Validation:** Mulugeta Dawit, Meareg Amare.

**Writing – original draft:** Mulugeta Dawit.

**Writing – review & editing:** Meareg Amare.

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
