## [Decision Letter · Decision Letter 0]

16 Oct 2020

PONE-D-20-29971

Poly(alizarin red S) modified glassy carbon electrode for square wave adsorptive stripping voltammetric determination of metronidazole in tablet formulation

PLOS ONE

Dear Dr. Amare,

Thank you for submitting your manuscript to PLOS ONE. After careful consideration, we feel that it has merit but does not fully meet PLOS ONE’s publication criteria as it currently stands. Therefore, we invite you to submit a revised version of the manuscript that addresses the points raised during the review process.

Based on reviewers recommendations, the manuscript required extensive revision before consider it for the publication in journal.

We look forward to receiving your revised manuscript.

Kind regards,

Girish Sailor

Academic Editor

PLOS ONE

Journal Requirements:

3.Thank you for stating the following financial disclosure:

 [The funders had no role in study design, data collection and analysis, decision to publish, or preparation of the manuscript.].

4. Please ensure that you refer to Figure 12 and 13 in your text as, if accepted, production will need this reference to link the reader to the figure.

Reviewers' comments:

Reviewer's Responses to Questions

**Comments to the Author**

1. Is the manuscript technically sound, and do the data support the conclusions?

Reviewer #1: Partly

Reviewer #2: Yes

Reviewer #3: Yes

2. Has the statistical analysis been performed appropriately and rigorously? 

Reviewer #1: No

Reviewer #2: Yes

Reviewer #3: Yes

3. Have the authors made all data underlying the findings in their manuscript fully available?

Reviewer #1: Yes

Reviewer #2: Yes

Reviewer #3: Yes

4. Is the manuscript presented in an intelligible fashion and written in standard English?

Reviewer #1: No

Reviewer #2: Yes

Reviewer #3: No

5. Review Comments to the Author

Reviewer #1: Review Report on “Poly (alizarin red S) modified glassy carbon electrode for square wave adsorptive stripping voltammetric determination of metronidazole in tablet formulation”

Manuscript Number: PONE-D-20-29971

The authors have given detailed information for the detection of metronidazole by glassy carbon electrode modified with electro-polymerized alizarin red S. The work lacks novelty and needs more issues to be addressed as follows:

A. ABSTRACT

1. Typo-errors should be adjusted along the whole manuscript.

2. SEM images of bare and unmodified GCE should be added.

3. R2 should be four decimal points.

4. Simple abbreviations should be used along the manuscript such as current should be Ipc, scan rate should be ν and ---------so on.

5. Recoveries % should be one decimal points for example 96.15% should be 96.2%

6. Line 20, in the presence of 100 and 150% of uric acid and ascorbic acid. Please omit these values and replace by concentrations.

7. What is meant by complex matrix? Why the authors did not quantify metronidazole in biological matrices such as urine and plasma.

8. What is meant by charge transfer controlled?

B. INTRODUCTION

9. Scheme 1 should be removed. The structure of metronidazole is already described in the reduction mechanism.

10. Metronidazole should be properly abbreviated as MTZ along the manuscript.

11. Thus, development of a simple, cost effective, and sensitive method for determination of MTZ in samples like tablet formulation is still vital. Please omit “like tablet formulation”.

12. The authors should add a simple paragraph about the importance of polymers in the fabrication of electrodes, and the following papers for references that force this section:

https://doi.org/10.1016/j.msec.2017.02.092, https://doi.org/10.1007/s00216-019-02245-8, https://doi.org/10.1002/elan.201700078, https://doi.org/10.1016/j.colsurfa.2019.01.033

13. Lines 73-75 not correct and should be corrected i.e. what is meant by glassy carbon electrode is widely used due to its biocompatibility with tissue (incorrect statement) and minimal propensity to show deteriorated response as a result of electrode fouling (not correct as the GCE itself cannot prevent fouling itself but the some modifier on its surface prevents its fouling).

III. EXPERIMENTAL

14. Abbreviations such as PARS and ARS should be defined at start i.e. ABSTRACT.

15. Term MET instead of metronidazole should be unified along the revised manuscript.

16. Concentration of supporting electrolyte in reference electrode should be mentioned.

17. Section 2.5, 10 mM should be replaced by amount in grams.

18. The authors should perform MET analysis in more interfering samples such as urine and plasma.

IV. RESULTS AND DISCUSSIONS

19. In Fig.1, to prove the stability of poly (alizarin S) on the surface of GCE, multiple CV scans should be provided on inset of Fig.1.

20. Accumulation time should be added to Fig.2.

21. Surface areas of bare and modified GCE should be calculated.

22. Scan rate should be unified along the manuscript either mV/s or mV s-1.

23. Figs. 5 and 6 should be merged in one Figure.

24. Line 257, R2 is determination coefficient not correlation coefficient.

25. What is the relationship between R2 and type of analyte transfer during electroanalysis (lines 257-260?

26. “Slope value of 0.38, which is less than the ideal value of 0.5 for plot of log Ip versus log ʋ (Fig. 9), further confirmed participation of both with adsorption as the predominant”. This is incorrect as slope near to 0.5 means diffusion controlled process while near to 1.0 means adsorption controlled. Please, use these references as a guide.

https://doi.org/10.1016/j.bios.2018.03.015, https://doi.org/10.1016/j.molliq.2018.08.105, https://doi.org/10.1016/j.bios.2019.111849

27. Fig.8, what is meant by SQRT?

28. Figs. 7, 8 and 9 should be merged in one Figure.

29. Accumulation time should be added to Fig.10.

30. Fig.11 represents calibration plot of MTZ in bulk or tablet. Please, specify.

31. Please, replace SWAdCSV by SWV.

32. Method should be compared with a reference method.

33. Why the authors tested selectivity of the electrode using only ascorbic and uric acids, although they performed their analysis if pharmaceutical formulations that include other excepients such as glucose, sucrose, Mg-stearate, maltose, starch and others. Ascorbic and uric acids are substances mainly present in biological samples. So, it is preferred to carry out analysis in biological fluid e.g. plasma

34. Figs. 12 and 13, replace SWSV by SWV.

35. Caption of Table 3, replace metronidazole with MTZ.

36. To Table 3, add samples analyzed.

37. Table 3, replace DPSV by DPV.

V. CONCLUSION (S)

38. Conclusion part should be shortened and contains all corrected issues raised by the reviewer.

39. Replace metronidazole by MTZ in conclusion part.

Reviewer #2: The results presented by the authors are important to understand a novel way to determine metroimidazole by voltammetry method using a modified electrode which is quite important. The findings are interesting. There are some minor revisions upon.

1. Please read MS carefully there are some minor grammar mistakes.

2. Insert the error bars in Figure 6 and 11.

3. The conclusion should be concise and highlighted the obtained results.

Reviewer #3: The manuscript introduced a stripping voltammetric method for the determination of

metronidazole in tablet sample based on the poly(alizarin red s) modified GCE. Although the method is simple, however, the author has not pointed out the advantages of it, when compared with other electrochemical methods for the determination of metronidazole. And the following questions should be considered by the author：

1. In line 106 and 107, “working” and “reference” can not be fully expressed, it should be replaced by professional terms such as working electrode, reference electrode.

2. The morphology of poly alizarin red material should be characterized by SEM if possible.

3. Some typing error should be corrected.

4. Table 2 should be represented by a three line table.

5. For the explanation of inner probes, it is better to compare the electrochemical behavior of outer probes such as ruthenium ammonia. In addition, some References is need to explanation the inner probes. For example, the article of Richard L. McCreery, https://pubs.acs.org/doi/abs/10.1021/ac960492r.

6. Error bars should be added to the calibration curve of Ic to c.

6. PLOS authors have the option to publish the peer review history of their article (what does this mean?). If published, this will include your full peer review and any attached files.

Reviewer #1: **Yes: **Dr. Mohamed M. El-Wekil

Reviewer #2: No

Reviewer #3: No

---

## [Author Response · Author response to Decision Letter 0]

22 Oct 2020

Response to comments by the reviewers is attached as a file (attach files).

---

## [Decision Letter · Decision Letter 1]

25 Nov 2020

PONE-D-20-29971R1

Poly(alizarin red S) modified glassy carbon electrode for square wave adsorptive stripping voltammetric determination of metronidazole in tablet formulation

PLOS ONE

Dear Dr. Amare,

Thank you for submitting your manuscript to PLOS ONE. After careful consideration, we feel that it has merit but does not fully meet PLOS ONE’s publication criteria as it currently stands. Therefore, we invite you to submit a revised version of the manuscript that addresses the points raised during the review process.

The reviewer has suggested certain minor correction in revised manuscript before publication. 

We look forward to receiving your revised manuscript.

Kind regards,

Girish Sailor

Academic Editor

PLOS ONE

Reviewers' comments:

Reviewer's Responses to Questions

**Comments to the Author**

1. If the authors have adequately addressed your comments raised in a previous round of review and you feel that this manuscript is now acceptable for publication, you may indicate that here to bypass the “Comments to the Author” section, enter your conflict of interest statement in the “Confidential to Editor” section, and submit your "Accept" recommendation.

Reviewer #1: (No Response)

Reviewer #3: (No Response)

2. Is the manuscript technically sound, and do the data support the conclusions?

Reviewer #1: Partly

Reviewer #3: Yes

3. Has the statistical analysis been performed appropriately and rigorously? 

Reviewer #1: Yes

Reviewer #3: Yes

4. Have the authors made all data underlying the findings in their manuscript fully available?

Reviewer #1: Yes

Reviewer #3: Yes

5. Is the manuscript presented in an intelligible fashion and written in standard English?

Reviewer #1: Yes

Reviewer #3: Yes

6. Review Comments to the Author

Reviewer #1: Manuscript Number: PONE-D-20-29971 R1

The authors have given detailed information for the detection of metronidazole by glassy carbon electrode modified with electro-polymerized alizarin red S. The work can be accepted after addressing the following:

In General: the abbreviations along the revised version should be used properly such as metronidazole should be MTZ, cyclic voltammetry should be CV and square wave voltammetry----etc.

A. ABSTRACT

1. Lines 9-10, with three-folds of current enhancement at the modified GCE. Compared to what?

2. Lines 13-15, this sentence should be adjusted.

3. Lines 20 and 21, concentration units of ascorbic and uric acids should be added.

4. Line 25, with complex matrix should be removed. Most tablets contain electro-inactive species, while biological samples contain more interfering species.

B. INTRODUCTION

5. Line 46, scheme1 should be removed.

6. The novelty of the manuscript is still unclear. The authors should force the readers’ attention via stressing on the novelty of the method compared to others.

B. EXPERIMENTAL

7. Internal diameter of GCE should be added.

8. Line 126, 5 mM should be 5.0 mM.

9. Line 127, 1mM should be 1.0 mM.

B. RESULTS AND DISCUSSIONS

10. Calculation of effective surface areas of GCE and modified GCE should be added.

11. Reproducibility and stability of PARS/GCE should be addressed.

12. To show accuracy of the method, it should be compared with reported method.

Reviewer #3: The manuscript introduced a stripping voltammetric method for the determination of

metronidazole in tablet sample based on the poly(alizarin red s) modified GCE. Some minor problems still exist and the author should examine the manuscript carefully and revise it：

1. In line 110 in the PONE-D-20-29971-R1, “Ag/AgCl (3 M KCl) as reference” should be replaced by professional terms such as reference electrode.

2. Table 2 still needs to be modified and it should be on the same page. The form of the table 2 is as shown in Table 3.

3. Incorect sentence and typing error still exist. Please check it carefully and correct it. For example, “0.98830 and 0.97398” in line 261, “[32 33.” in line 264 in the PONE-D-20-29971-R1.

4. Some superscripts and subscripts should be carefully checked and revised.

7. PLOS authors have the option to publish the peer review history of their article (what does this mean?). If published, this will include your full peer review and any attached files.

Reviewer #1: No

Reviewer #3: No

---

## [Author Response · Author response to Decision Letter 1]

25 Nov 2020

Response to comments by reviewers

Firstly, we authors appreciate the reviewers for their critical comments. Below is our point-by-point response to comments by the reviewers.

Reviewer #1

General comment: 

In General: the abbreviations along the revised version should be used properly such as metronidazole should be MTZ, cyclic voltammetry should be CV and square wave voltammetry----etc.

Response

• Thanks to the reviewer, we have revised the document addressing the comment.

ABSTRACT

1. Lines 9-10, with three-folds of current enhancement at the modified GCE. Compared to what?

Answer: addressed in the revised document.

2. Lines 13-15, this sentence should be adjusted.

Answer: the sentence is rewritten.

3. Lines 20 and 21, concentration units of ascorbic and uric acids should be added.

Answer: common unit is added.

4. Line 25, with complex matrix should be removed. Most tablets contain electro-inactive species, while biological samples contain more interfering species.

Answer: In electrochemistry, interferent does not only refer to an electroactive species but also to any form of substance that may alter the result may be by undergoing a reaction with the analyte of interest or by competing the potential space of the analyte. Thus, by a complex matrix in our context refers to any including the excipient. Thus, the comment is not accepted.

INTRODUCTION

5. Line 46, scheme1 should be removed.

Answer: the reason why it is customary to put the structure of the analyte in the document is it helps the reader to predict (ask himself) the possible reaction sites and reaction type the analyte could undergo. Therefore, the authors are not convinced why to remove the scheme.

6. The novelty of the manuscript is still unclear. The authors should force the readers’ attention via stressing on the novelty of the method compared to others.

Answer: Thanking the reviewer for his/her critical comment, we have tried to modify the last statement of the introduction part addressing the comment. 

EXPERIMENTAL

7. Internal diameter of GCE should be added.

Answer: the internal diameter of the GCE used in this study is already mentioned under section 2.3.

8. Line 126, 5 mM should be 5.0 mM.

Answer: Addressed.

9. Line 127, 1mM should be 1.0 mM.

Answer: Addressed.

RESULTS AND DISCUSSIONS

10. Calculation of effective surface areas of GCE and modified GCE should be added.

Answer: The comment to determine effective surface area among the two possible contributors for catalytic activity (conductivity, and electrode effective surface area) is highly appreciated. Although we have shown the conductivity improvement of the modified electrode, we failed to have data of the effect of scan rate on [Fe(CN)6]3-/4-. Appreciating the comment, we couldn’t include the data in the revised document. 

11. Reproducibility and stability of PARS/GCE should be addressed.

Answer: For the absolute completeness of our work, we should have validated it using parameters including the reproducibility and stability of the modified electrode. Appreciating the reviewer’s comment, we failed to do it for we are out of lab for CORONA reason. 

12. To show accuracy of the method, it should be compared with reported method.

Answer: the performance of our method is compared with the nominal tablet labeled value and further validated by the recovery (spike and interference) results. 

Reviewer #3

1. In line 110 in the PONE-D-20-29971-R1, “Ag/AgCl (3 M KCl) as reference” should be replaced by professional terms such as reference electrode.

Answer: To avoid confusions between the different reference electrode like calomel, standard hydrogen electrode, Ag/AgCl (saturated,…), it is advisable to put the specific electrode used.

2. Table 2 still needs to be modified and it should be on the same page. The form of the table 2 is as shown in Table 3.

Answer: Thanking the comment by the reviewer, we have polished the table (removed the raw lines).

3. Incorrect sentence and typing error still exist. Please check it carefully and correct it. For example, “0.98830 and 0.97398” in line 261, “[32 33.” in line 264 in the PONE-D-20-29971-R1.

Answer: we have revised the document addressing typographic errors including what the reviewer as indicated. However, the “0.98830” and “0.97398” which are the calculated determination coefficients are important values based on which we proposed an adsorption reaction kinetics.

4. Some superscripts and subscripts should be carefully checked and revised.

Answer: We have revised the document addressing such errors.

---

## [Decision Letter · Decision Letter 2]

3 Dec 2020

Poly(alizarin red S) modified glassy carbon electrode for square wave adsorptive stripping voltammetric determination of metronidazole in tablet formulation

PONE-D-20-29971R2

Dear Dr. Amare,

We’re pleased to inform you that your manuscript has been judged scientifically suitable for publication and will be formally accepted for publication once it meets all outstanding technical requirements.

Kind regards,

Girish Sailor

Academic Editor

PLOS ONE

Additional Editor Comments (optional):

Reviewers' comments:

Reviewer's Responses to Questions

**Comments to the Author**

1. If the authors have adequately addressed your comments raised in a previous round of review and you feel that this manuscript is now acceptable for publication, you may indicate that here to bypass the “Comments to the Author” section, enter your conflict of interest statement in the “Confidential to Editor” section, and submit your "Accept" recommendation.

Reviewer #1: (No Response)

Reviewer #3: (No Response)

2. Is the manuscript technically sound, and do the data support the conclusions?

Reviewer #1: (No Response)

Reviewer #3: (No Response)

3. Has the statistical analysis been performed appropriately and rigorously? 

Reviewer #1: Yes

Reviewer #3: (No Response)

4. Have the authors made all data underlying the findings in their manuscript fully available?

Reviewer #1: Yes

Reviewer #3: (No Response)

5. Is the manuscript presented in an intelligible fashion and written in standard English?

Reviewer #1: Yes

Reviewer #3: (No Response)

6. Review Comments to the Author

Reviewer #1: (No Response)

Reviewer #3: (No Response)

7. PLOS authors have the option to publish the peer review history of their article (what does this mean?). If published, this will include your full peer review and any attached files.

Reviewer #1: No

Reviewer #3: No

---

## [Editor Report · Acceptance letter]

7 Dec 2020

PONE-D-20-29971R2 

Poly(alizarin red S) modified glassy carbon electrode for square wave adsorptive stripping voltammetric determination of metronidazole in tablet formulation 

Dear Dr. Amare:

I'm pleased to inform you that your manuscript has been deemed suitable for publication in PLOS ONE. Congratulations! Your manuscript is now with our production department. 

Kind regards, 

on behalf of

Dr. Girish Sailor 

Academic Editor

PLOS ONE